

# Charge-current correlation equalities
# for quantum systems far from equilibrium

**Dragi Karevski[1] and Gunter M. Schütz[2⋆]**

**1** Laboratoire de Physique et Chimie Théoriques, Université de Lorraine,
UMR CNRS 7019, B.P. 70239, F-54506 Vandoeuvre les Nancy Cedex, France
**2** Institute of Complex Systems II, Theoretical Soft Matter and Biophysics,
Forschungszentrum Jülich, 52425 Jülich, Germany

⋆ g.schuetz@fz-juelich.de

## Abstract

We prove that a recently derived correlation equality between conserved charges and
their associated conserved currents for quantum systems far from equilibrium [O.A.
Castro-Alvaredo, B. Doyon, and T. Yoshimura, Phys. Rev. X 6, 041065 (2016)], is valid
under more general conditions than assumed so far. Similar correlation identities, which
in generalized Gibbs ensembles give rise to a current symmetry somewhat reminiscent
of the Onsager relations, turn out to hold also in the absence of translation invariance,
for lattice models, and in any space dimension, and to imply a symmetry of the non-
equilibrium linear response functions.


# 1 Introduction

Of particular interest in the general context of transport far from thermal equilibrium are the correlations between the conserved charges $Q^\alpha$ and their associated currents $J_i^\alpha$ in space direction $i$. We refer to the review by Spohn [1] for a discussion from a broad perspective. Very recently, charge-current correlations in one-dimensional quantum integrable systems have been shown to play an important role in work on the Drude weight [2,3] and for generalized hydrodynamics [4,5].

Specifically, for the one-dimensional quantum case the global charge-current symmetry

$$\langle Q^\alpha J^\beta \rangle^c = \langle J^\alpha Q^\beta \rangle^c \tag{1}$$

for the connected correlation functions has been derived in [4] under quite general circumstances, viz., assuming only translation invariance of the stationary density matrix and the quantum Hamiltonian, a generic assumption on the decay of correlations, and, more significantly, commutativity of the stationary density matrix with the charges $Q^\alpha$.

This result was subsequently generalized to a stronger local version $\langle q^\alpha(x,t) j^\beta(0,0) \rangle^c = \langle j^\alpha(x,t) q^\beta(0,0) \rangle^c$ [6] which does *not* require the assumption of commutativity of the charges and which is valid for any decay of correlations with distance. The main aim of the present work is to derive related global and local charge-current correlation equalities and to clarify the necessary and sufficient conditions under which such correlation equalities, including (1) and its local version, are valid.

We start out from generic stationary one-dimensional lattice quantum systems of finite size with local conservation laws, without requiring translation invariance, as detailed in Sec. 2. The main results are derived in Sec. 3, first very generally and then more specifically under various additional generic conditions imposed on the physical system. Some simple consequences for symmetries of far-from-equilibrium linear response functions are indicated in Sec. 4.

All results derived below are straightforwardly extended to higher dimensions by projection on one space coordinate and going through the same steps as below for each space coordinate. Furthermore, the results are valid also for dissipative quantum systems where the time evolution of the density matrix is generated by a Lindblad quantum master equation, and for purely classical stochastic systems with Markovian dynamics for the probability distribution. However, to avoid heavy notation and to expose clearly the essential ingredients that lead to charge-current correlation equalities, we stick to the one-dimensional quantum context.

# 2 The setting

We consider a stationary many-body quantum system on a one-dimensional lattice of $L$ sites with Hamiltonian $H$. We shall not from the outset assume translation invariance, but allow for non-translation-invariant stationary density matrices $\rho$ and/or spatially inhomogeneous dynamics encoded in $H$. The system is not assumed to be in thermal equilibrium. Stationarity only means that we take expectations w.r.t. a density matrix $\rho$ that satisfies

$$\textbf{S1:} \qquad [\rho, H] = 0. \tag{2}$$

For observables $O$ we recall the definition

$$O(t) = e^{\frac{i}{\hbar}Ht} O e^{-\frac{i}{\hbar}Ht} \tag{3}$$

of time-dependent operators in the Heisenberg picture. We denote stationary expectation values and connected correlation functions by $\langle O \rangle_L := \text{tr}(\rho O)$ and

$\langle O_1(t)O_2\rangle_L^c := \mathrm{tr}(\rho O_1(t)O_2) - \langle O_1\rangle\langle O_2\rangle$ resp., with the size-dependence indicated by the subscript $L$.

Specifically, we consider a family of $n$ locally conserved charges, i.e., operators $q_k^\alpha$ that satisfy for $k \in \{1,\ldots,L\}$ the discrete continuity equation

$$\mathbf{S2:} \qquad \frac{i}{\hbar}\big[H, q_k^\alpha(t)\big] = j_{k-1}^\alpha(t) - j_k^\alpha(t), \tag{4}$$

with the conserved currents $j_k^\alpha$ and the definition $j_0^\alpha := j_L^\alpha$. Then the operators

$$Q^\alpha = \sum_k q_k^\alpha \tag{5}$$

form a set of $n$ conserved charges $Q^\alpha = Q^\alpha(t)$. We remark that for a non-translation invariant $H$ the operator $j_{k-1}^\alpha(t)$ may not be the lattice translation of $j_k^\alpha(t)$ but may have an explicit $k$-dependence that we do not indicate. Nevertheless, the discrete continuity equation (4) alone implies that the stationary current, denoted by $j^\alpha$, does not depend on $k$.

It is tacitly assumed that the charge and current operators are bounded so that all stationary expectations of the charges $q_k^\alpha(t)$ and currents $j_l^\alpha(t)$ and all stationary correlations between them are finite for all system sizes $L$ and have well-defined thermodynamic limits.

In $d > 1$ dimensions the lattice continuity equation for the locally conserved charges $q_{\mathbf{k}}^\alpha(t)$ at the lattice point $\mathbf{k} = (k_1,\ldots,k_d)$ reads

$$\frac{i}{\hbar}\big[H, q_{\mathbf{k}}^\alpha(t)\big] = \sum_{i=1}^{d}\Big[j_{\mathbf{k}_i^-}^{i,\alpha}(t) - j_{\mathbf{k}}^{i,\alpha}(t)\Big], \tag{6}$$

with the conserved currents $j_{\mathbf{k}}^{i,\alpha}(t)$ in space direction $i$ and the shifted $i^{th}$ coordinate $\mathbf{k}_i^- := (k_1,\ldots,k_i-1,\ldots,k_d)$. One considers the projected operators

$$q_{k_i}^\alpha(t) = \sum_{\mathbf{k}\backslash k_i} q_{\mathbf{k}}^\alpha(t), \tag{7}$$

$$j_{k_i}^{i,\alpha}(t) = \sum_{\mathbf{k}\backslash k_i} j_{\mathbf{k}}^{i,\alpha}(t), \tag{8}$$

where the summations exclude the space coordinate $i$ and goes through the same calculations as below for the one-dimensional case.

## 3 Charge-current correlation equalities

Specifically, we consider the time-dependent stationary correlation functions

$$S_L^{\alpha\beta}(k,l,t) := \langle q_k^\alpha(t)q_l^\beta(0)\rangle_L^c, \tag{9}$$

$$C_L^{\alpha\beta}(k,l,t) := \langle j_k^\alpha(t)q_l^\beta(0)\rangle_L^c, \tag{10}$$

$$\tilde{C}_L^{\alpha\beta}(k,l,t) := \langle q_k^\alpha(t)j_l^\beta(0)\rangle_L^c. \tag{11}$$

By identifying all lattice sites $k$ modulo $L$, the correlation functions can be defined for all $k,l \in \mathbb{Z}$ with periodicity $L$ for both space arguments $k,l$.

### 3.1 Results of general validity

In this subsection we study relations between the charge-current correlation functions (10) and (11) that arise alone from S1 and S2, i.e., stationarity of the density matrix (2) and the conservation law (4), without requiring translation invariance or any other specific property of $\rho$ or $H$.

(i) With the Heisenberg representation (3) and the cyclic invariance of the trace, one gets for the time derivative of the charge-charge correlation function (9) the two expressions

$$
\begin{aligned}
\dot{S}_L^{\alpha\beta}(k,l,t) &= \langle (j_{k-1}^\alpha(t) - j_k^\alpha(t)) q_l^\beta(0) \rangle_L^c & (12) \\
&= -\langle q_k^\alpha(t)(j_{l-1}^\beta(0) - j_l^\beta(0)) \rangle_L^c, & (13)
\end{aligned}
$$

from which one deduces by subtraction the fundamental charge-current correlation equality

$$
0 = C_L^{\alpha\beta}(k-1,l,t) - C_L^{\alpha\beta}(k,l,t) + \tilde{C}_L^{\alpha\beta}(k,l-1,t) - \tilde{C}_L^{\alpha\beta}(k,l,t). \tag{14}
$$

which is local in both coordinates $k$ and $l$ and which is the basis for further considerations.

(ii) To explore consequences of this relation we consider the correlations involving the total charges $Q^\alpha$, viz.,

$$
\begin{aligned}
A_L^{\alpha\beta}(k,t) &:= \sum_l C_L^{\alpha\beta}(k,l,t), & (15) \\
\tilde{A}_L^{\alpha\beta}(l,t) &:= \sum_k \tilde{C}_L^{\alpha\beta}(k,l,t). & (16)
\end{aligned}
$$

Because of the global charge conservation (5), both averages $A_L^{\alpha\beta}(k,t)$ and $\tilde{A}_L^{\alpha\beta}(l,t)$ are trivially independent of time. The local relations (12) and (13) then imply that both functions are independent also of the space coordinate. This yields without further computation the charge-current correlation equalities

$$
\begin{aligned}
A_L^{\alpha\beta}(k,t) &= \langle j_k^\alpha(t) Q^\beta \rangle_L^c = \langle j_0^\alpha(0) Q^\beta \rangle_L^c =: a_L^{\alpha\beta}, & (17) \\
\tilde{A}_L^{\alpha\beta}(l,t) &= \langle Q^\alpha(t) j_l^\beta(0) \rangle_L^c = \langle Q^\alpha j_0^\beta(0) \rangle_L^c =: \tilde{a}_L^{\alpha\beta}, & (18)
\end{aligned}
$$

with constants $a_L^{\alpha\beta}$, $\tilde{a}_L^{\alpha\beta}$ that depend neither on $k$ nor on $t$.

(iii) Next we consider the space averages

$$
\begin{aligned}
B_L^{\alpha\beta}(r,t) &:= \frac{1}{L} \sum_k C_L^{\alpha\beta}(k, k+r, t), & (19) \\
\tilde{B}_L^{\alpha\beta}(r,t) &:= \frac{1}{L} \sum_k \tilde{C}_L^{\alpha\beta}(k, k+r, t). & (20)
\end{aligned}
$$

For examining the relationship between $B_L^{\alpha\beta}(k,t)$ and $\tilde{B}_L^{\alpha\beta}(l,t)$ we define the auxiliary function

$$
G_L^{\alpha\beta}(k,l,t) := \sum_{k'=1}^{k} \left[ \tilde{C}_L^{\alpha\beta}(k',0,t) - \tilde{C}_L^{\alpha\beta}(k',l,t) \right] \tag{21}
$$

and its space average

$$
g_L^{\alpha\beta}(r,t) := \frac{1}{L} \sum_k G_L^{\alpha\beta}(k, k+r, t), \tag{22}
$$

which allow for expressing both $C_L^{\alpha\beta}(k,t)$ and $\tilde{C}_L^{\alpha\beta}(l,t)$ in terms of $G_L^{\alpha\beta}(k,l,t)$ and the space averages $B_L^{\alpha\beta}(k,t)$ and $\tilde{B}_L^{\alpha\beta}(l,t)$ in terms of $g_L^{\alpha\beta}(r,t)$. The auxiliary function $G_L^{\alpha\beta}(k,l,t)$ satisfies $G_L^{\alpha\beta}(k,0,t) = G_L^{\alpha\beta}(0,l,t) = 0$ and the periodicity property

$$G_L^{\alpha\beta}(k+mL,l+nL,t) = G_L^{\alpha\beta}(k,l,t) \tag{23}$$

that is inherited from the periodicity of the correlation functions. Similarly, one has $g_L^{\alpha\beta}(r+mL,t) = g_L^{\alpha\beta}(r,t)$.

One gets from the definition (21) and from the doubly local relation (14)

$$C_L^{\alpha\beta}(k,l,t) = C_L^{\alpha\beta}(0,l,t) + G_L^{\alpha\beta}(k,l,t) - G_L^{\alpha\beta}(k,l-1,t) \tag{24}$$

$$\tilde{C}_L^{\alpha\beta}(k,l,t) = \tilde{C}_L^{\alpha\beta}(k,0,t), + G_L^{\alpha\beta}(k-1,l,t) - G_L^{\alpha\beta}(k,l,t). \tag{25}$$

By setting $l = k+r$ in (24) and (25) and summing over $k$ one finds from the charge-current correlation equalities (17) and (18)

$$B_L^{\alpha\beta}(r,t) = \frac{1}{L}a_L^{\alpha\beta} + g_L^{\alpha\beta}(r,t) - g_L^{\alpha\beta}(r-1,t), \tag{26}$$

$$\tilde{B}_L^{\alpha\beta}(r-1,t) = \frac{1}{L}\tilde{a}_L^{\alpha\beta} + g_L^{\alpha\beta}(r,t) - g_L^{\alpha\beta}(r-1,t) \tag{27}$$

in terms of the space average (22). Thus we arrive at the charge-current correlation equality

$$B_L^{\alpha\beta}(r+1,t) - \tilde{B}_L^{\alpha\beta}(r,t) = \frac{1}{L}\alpha_L^{\alpha\beta} \quad \forall r,t, \tag{28}$$

with the constant $\alpha_L^{\alpha\beta} := a_L^{\alpha\beta} - \tilde{a}_L^{\alpha\beta}$.

The constant $\alpha_L^{\alpha\beta}$ is given by (17) and (18)

$$\alpha_L^{\alpha\beta} = \langle j_0^\alpha(0)Q^\beta \rangle_L^c - \langle Q^\alpha j_0^\beta(0) \rangle_L^c \tag{29}$$

in terms of the stationary charge-current correlations for the global charges. Notice that the independence of $r$ and $t$ allows for expressing $\alpha_L^{\alpha\beta}$ also as a stationary long-distance correlation as

$$\alpha_L^{\alpha\beta} = L[B_L^{\alpha\beta}(\lfloor L/2 \rfloor + 1,0) - \tilde{B}_L^{\alpha\beta}(\lfloor L/2 \rfloor,0)], \tag{30}$$

where $\lfloor x \rfloor \in \mathbb{Z}$ is the integer part of $x \in \mathbb{R}$. This is a finite-size term that is generically small, but can be relevant for long-range interactions or non-local conserved charges. Also in the presence of stationary long-range correlations at or below a quantum critical point the correlation may not be negligible.

As an aside we note without further comment that by (12) and (13) the auxiliary function $G^{\alpha\beta}(k,l,t)$ is related to the structure function as

$$\dot{S}_L^{\alpha\beta}(k,l,t) = G_L^{\alpha\beta}(k-1,l,t) + G_L^{\alpha\beta}(k,l-1,t)$$
$$- G_L^{\alpha\beta}(k-1,l-1,t) - G_L^{\alpha\beta}(k,l,t). \tag{31}$$

For the space average

$$s_L^{\alpha\beta}(r,t) := \frac{1}{L}\sum_k S_L^{\alpha\beta}(k,k+r,t), \tag{32}$$

one gets the evolution equation

$$\dot{s}_L^{\alpha\beta}(r,t) = g_L^{\alpha\beta}(r+1,t) + g_L^{\alpha\beta}(r-1,t) - 2g_L^{\alpha\beta}(r,t). \tag{33}$$

## 3.2 Specializations

The results (14), (17), (18), and (28) - (30) are valid without any conditions on the density matrix $\rho$ and on the Hamiltonian $H$, except that all correlations are assumed to be bounded. Now we consider some conditions of a general character and explore their consequences.

### 3.2.1 Decay of correlations

We make the generic assumption of decay of correlations in the thermodynamic limit $L \to \infty$, i.e., for all $r, t$ we postulate

$$\textbf{C1:} \quad \lim_{r \to \infty} B_\infty^{\alpha\beta}(r, t) = \lim_{r \to \infty} \tilde{B}_\infty^{\alpha\beta}(r, t) = 0. \tag{34}$$

This assumption is justified by the finite Lieb-Robinson speed in non-relativistic quantum mechanics [7].

Decay of correlations implies $\alpha_L^{\alpha\beta}/L \to 0$ for $L \to \infty$ and therefore (28) yields the asymptotic charge-current correlation equality

$$B_\infty^{\alpha\beta}(r + 1, t) = \tilde{B}_\infty^{\alpha\beta}(r, t) \tag{35}$$

for the space averaged correlation function.

Under the slightly stronger condition

$$\textbf{C1':} \lim_{L \to \infty} L[B_L^{\alpha\beta}(\lfloor L/2 \rfloor + 1, 0) - \tilde{B}_L^{\alpha\beta}(\lfloor L/2 \rfloor, 0)] = 0 \tag{36}$$

on the decay of correlations one has $\alpha_L^{\alpha\beta} \to 0$ for $L \to \infty$. Then (29) yields

$$\langle j_0^\alpha(0) Q^\beta \rangle_\infty^c = \langle Q^\alpha j_0^\beta(0) \rangle_\infty^c. \tag{37}$$

We stress that no translation invariance is used to prove (35) and (37).

### 3.2.2 Translation invariance

Now we consider the case where both $\rho$ and $H$ are translation invariant, i.e., for the lattice translation operator $T$ that transforms observables indexed by site $k$ into the same observable for site $k + 1 \pmod L$ one has

$$\textbf{C2:} \quad T\rho T^{-1} = \rho, \quad THT^{-1} = H. \tag{38}$$

Then $B_L^{\alpha\beta}(r, t) = C_L^{\alpha\beta}(0, r, t)$ and $\tilde{B}_L^{\alpha\beta}(r, t) = \tilde{C}_L^{\alpha\beta}(0, r, t)$ and (28) becomes

$$\langle j_k^\alpha(t) q_{l+1}^\beta(0) \rangle_L^c - \langle q_k^\alpha(t) j_l^\beta(0) \rangle_L^c = \frac{1}{L}\alpha_L^{\alpha\beta}, \tag{39}$$

with the constant $\alpha_L^{\alpha\beta}$ given in (29).

We note that condition C2 together with C1 (decay of correlations) yields

$$\langle j_k^\alpha(t) q_0^\beta(0) \rangle_\infty^c = \langle q_{k+1}^\alpha(t) j_0^\beta(0) \rangle_\infty^c, \tag{40}$$

which is the lattice analogue of the local charge-current correlation equality derived for translation invariant systems in continuous space in [6].

### 3.2.3 Mutually commuting charges

We finally comment on mutually commuting charges where

$$\textbf{C3:} \qquad [Q^\alpha, Q^\beta] = [Q^\alpha, \rho] = 0 \tag{41}$$

for a set of charges labelled by $\alpha, \beta$.

(1) First we consider a canonical ensemble where the density matrix is build from eigenstates of the conserved charges $Q^\alpha$ and $Q^\beta$, i.e., $Q^{\alpha,\beta}\rho = \rho Q^{\alpha,\beta} = Lq^{\alpha,\beta}\rho$ with the charge densities $q^{\alpha,\beta}$. Then $a_L^{\alpha\beta} = \tilde{a}_L^{\alpha\beta} = 0$ and (28) yields

$$B_L^{\alpha\beta}(r+1, t) = \tilde{B}_L^{\alpha\beta}(r, t) \tag{42}$$

for all $r$ and $t$ and any finite $L$, without assuming decay of correlations or translation invariance.

(2) Second, we consider a generalized Gibbs ensemble of the form

$$\tilde{\rho} = \frac{1}{Z}\rho e^{\sum_{\alpha=1}^n \lambda_\alpha Q^\alpha}, \tag{43}$$

with $Z = \text{tr}(\rho e^{\sum_{\alpha=1}^n \lambda_\alpha Q^\alpha})$ and stationary $\rho$ independent of the generalized chemical potentials $\lambda_\alpha$. It has been conjectured that such a GGE state emerges asymptotically in time when an integrable system, which has an extensive number of conserved local charges, has suffered a sudden quench [8, 9], see [10] for a general review. This conjecture has been checked explicitly in many non-interacting models, see for example [11, 12], and tested in truly interacting integrable models with a truncated GGE taking into account only a finite number $n$ of charges [13–18][1]. Indirect experimental evidence was found by Vidmar et al. [19] who confirmed the magnetization profile that was theoretically predicted for the evolution of the $XX$ quantum chain after a quench to a step initial state [20], see also [21] on the current fluctuations in this setting.

Given a GGE satisfying C3, which by construction (S1 and S2) is then also stationary, one has with the short-hand notation $\partial_\alpha \equiv \partial/(\partial \lambda_\alpha)$

$$\partial_\alpha \ln Z = \langle Q^\alpha \rangle, \quad \partial_\alpha \langle O \rangle = \langle OQ^\alpha \rangle^c. \tag{44}$$

Thus one can express the constants $a_L^{\alpha\beta}$ and $\tilde{a}_L^{\alpha\beta}$ as derivatives as

$$a_L^{\alpha\beta} = \partial_\beta j^\alpha, \quad \tilde{a}_L^{\alpha\beta} = \partial_\alpha j^\beta \tag{45}$$

and obtains from (30)

$$\partial_\beta j^\alpha - \partial_\alpha j^\beta = L[B_L^{\alpha\beta}(\lfloor L/2 \rfloor + 1, 0) - \tilde{B}_L^{\alpha\beta}(\lfloor L/2 \rfloor, 0)]. \tag{46}$$

We note that condition C3 for the GGE together with the condition C1' (36) on the decay of correlations yields the current symmetry

$$\partial_\beta j^\alpha = \partial_\alpha j^\beta, \tag{47}$$

where the stationary expectations $j^\alpha$ are understood as functions of the generalized chemical potentials $\lambda_\alpha$. No translation invariance is required.

The current symmetry (47) appears in many contexts in hydrodynamic theory, see e.g. [1, 22] for a review and [2, 4, 6] for recent applications in generalized hydrodynamics where it

---

[1]In some cases the set of local charges is not enough to specify the state of the system and one needs to extend the GGE by incorporating additional so called quasi-local charges, see [16–18]

was derived under the assumption C1' (decay of correlations in the form (36)), C2 (translation invariance) and C3 (GGE). A mathematically rigorous proof of this current symmetry in the classical Markovian context was presented earlier in [23], using the same conditions C1' and C2 and arguments for the proof that were later employed in similar form in [4, 22].

We also note that assumption C3 implies for the grandcanonical ensemble the relation $a_L^{\alpha\beta} = \tilde{a}_L^{\beta\alpha}$ and hence for $\alpha = \beta$ the exact charge-current correlation equality (42) for all $r$ and $t$ and any finite $L$.

# 4 Linear response symmetries

We point out some straightforward consequences of the charge-current correlation equalities for linear response in far-from-equilibrium systems. For definiteness, we assume conditions C1' (decay of correlations (36)) and C2 (translation invariance) to be satisfied.

Consider a time-dependent perturbation of the form $H(t) = H_0 + hA(t)$ where $h$ is the interaction strength. The linear-response function for an observable $B$ is given by [24]

$$\hat{R}_{AB}(t) := \frac{\mathrm{d}}{\mathrm{d}h} \langle B(t) \rangle \Big|_{h=0}. \tag{48}$$

For a pulse at time $t_0 = 0$, i.e., when the perturbation is of the form $A(t) = A\delta(t)$, and for a density matrix $\rho$ that is stationary under the evolution of $H_0$, straightforward computation yields $\hat{R}_{AB}(t) = R_{AB}(t)\Theta(t)$ where [24]

$$R_{AB}(t) = \frac{i}{\hbar} \mathrm{tr}\{\rho[A, B(t)]\}, \tag{49}$$

with the time-dependent operator $B(t)$.

Consider now the response at site $k$ of the observable $B = q_k^\beta$ to a pulse perturbation with $A = q_0^\alpha$ at the origin. Then (49) yields

$$R^{\alpha\beta}(k, t) = \frac{i}{\hbar} \mathrm{tr}\left\{\rho\left[q_0^\alpha, q_k^\beta(t)\right]\right\}. \tag{50}$$

The total response

$$R_0^{\alpha\beta} := \sum_k R^{\alpha\beta}(k, t) = \frac{i}{\hbar} \mathrm{tr}\left\{\rho\left[q_0^\alpha, Q^\beta\right]\right\} =: -R_0^{\beta\alpha} \tag{51}$$

is trivially antisymmetric in the indices and independent of time because of the conservation law.

Now consider the first moment

$$R_1^{\alpha\beta}(t) \quad := \sum_{k=\lfloor -L/2 \rfloor + 1}^{\lfloor L/2 \rfloor} k R^{\alpha\beta}(k, t), \tag{52}$$

which provides information about the position at time $t$ of the center of mass of the perturbation. Taking the time-derivative and using decay of correlations (36) yields a first moment

$$v^{\alpha\beta} := \dot{R}_1^{\alpha\beta}(t) = \langle \left[Q^\alpha, j^\beta(0)\right] \rangle_c \tag{53}$$

that does not depend on time so that $R_1^{\alpha\beta}(t) = R_1^{\alpha\beta}(0) + v^{\alpha\beta} t$ holds exactly.

Furthermore, from the global correlation equality (1) one derives the symmetry property

$$v^{\alpha\beta} = -v^{\beta\alpha} \tag{54}$$

between the first moments. Assuming further condition C3 (commutativity of the conserved charges with the stationary density matrix), one obtains

$$v^{\alpha\beta} = v^{\beta\alpha} = 0, \tag{55}$$

both in the canonical and grandcanonical ensemble.

## 5   Conclusions

The charge-current correlation equalities (14), (17), (18), and (28) - (30) are generally valid, without any specific hypothesis on the nature of a stationary quantum system with conserved charges $q_k^\alpha(t)$ that satisfy the discrete continuity equation (4) and have finite stationary cross correlations among themselves and with the currents $j_l^\alpha(t')$. More specialized equalities arise as when conditions C1 (34) or C1' (36) on the decay of correlations are assumed to hold (see (35), (37), and (54)) or if some of the conserved charges commute among themselves and with the stationary density matrix (see (42), (46), the current symmetry (47), and the linear response symmetry (55)). Translation invariance does not play a role for the validity of these correlation equalities.

These results clarify and generalize the range of validity of similar relations obtained in [4,6] for translation invariant systems. The correlation equalities are valid arbitrarily far from thermal equilibrium and provide concrete information about the spatial structure of the linear response function under these general conditions and about finite-size corrections involving the local charge-current correlations.

As pointed out in [25], the current symmetry (47) guarantees that for stationary GGE's only hyperbolic systems of conservation laws can arise as Eulerian hydrodynamic limits that govern the macroscopic time-evolution of the local conserved quantities. When the fluctuations of the locally conserved charges are the most relevant slow dynamical variables, one expects in one space dimension from mode-coupling theory [22] that fluctuations around the deterministic hydrodynamics are generically diffusive or in the Kardar-Parisi-Zhang (KPZ) universality class [26] and, on special manifolds in the space of densities and model parameters, in the Fibonacci universality classes [27] which include the diffusive and superdiffusive Kardar-Parisi-Zhang universality class as paradigmatic members. For recent evidence of diffusive and superdiffusive transport we mention [28,29] and more specifically on the observation of KPZ physics in the $SU(2)$-symmetric Heisenberg spin chain we refer to [30,31].

Finally, we note that the current symmetry (47) may be useful in numerical computations of quantum quenches as a probe of an underlying asymptotic GGE, as (47) would not be valid if the local stationary state does not approximate a GGE. Likewise, the linear response symmetry (55) can be used as a probe of the symmetries of a density matrix when its only *a priori* known property is stationarity.

## Acknowledgements

It is a pleasure to acknowledge stimulating discussions with A. Klümper and V. Popkov and to thank B. Doyon for helpful comments on an earlier version of this paper. This work was supported by Deutsche Forschungsgemeinschaft. G.M.S. thanks the Laboratoire de Physique et Chimie Théoriques, Université de Lorraine, where part of this work was done, for kind hospitality.

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
