# Peer review of "Charge-current correlation equalities for quantum systems far from equilibrium"

_SciPost Physics, doi:SciPost Phys. 6, 068 (2019)_

## Round 2 · Referee Report · Anonymous (Referee 1) · 2019-3-4

Strengths

  • Non-trivial and general results

Weaknesses

  • General presentation a bit rushed

Report

The authors show that the charge-current symmetry of equation 1 is valid on a generic density matrix (not necessarily commuting with all the conserved charges) and in any spacial dimension. The result of eq 1 was originally found in the context of Generalised Hydrodynamics but they show that it is actually more general. They give a fulfilling proof of their statement in any spacial dimension. The paper definitely deserves publication.

Requested changes

-I would request the authors to give a broader introduction of their results. Namely they could specify how eq 1 was useful in Generalised Hydrodynamics and which physical implications it had for integrable systems. - It seems to me that it should be made more clear that in Generalised Hydrodynamics eq 1 is a consequence of local equilibrium and therefore not a priori valid at short times. The authors instead claim that it is valid at any time, especially after some quench from an inhomogenous initial state. Is that correct? If yes it should be made more clear in the introduction. -In the introduction they could mention what are the assumption for their proof. It seems to me that one request is that the state has sufficiently rapid decay of correlations, eq 28. - below eq 34 they should define what is M and its physical meaning - slightly above eq 20: it seems that the authors want to point out that the symmetry eq 41 implies no hydrodynamics shocks. This point deserve to be expanded and made a bit more clear, also by referring to Generalised Hydrodynamics theory. - in the conclusion they authors says that the correlation equality may be used as a probe of of the underlying asymptotic GGE, but it is not clear how. - in the conclusion the authors says that fluctuations are expected to be described by the KPZ class. It has been recently show [Phys. Rev. Lett. 121, 160603] that fluctuations are typically diffusive and in some cases super-diffusive, see Phys. Rev. Lett. 121, 230602 and arXiv:1812.02701. The authors may refer to these references maybe for their claim.

  • validity: top
  • significance: high
  • originality: high
  • clarity: good
  • formatting: good
  • grammar: perfect

Author:  Gunter Schütz  on 2019-05-18  [id 521]

(in reply to Report 1 on 2019-03-04)

In response to the remarks by the referee we have rewritten significantly the introduction and rewritten completely the main part about charge-current correlations. In particular, we have structured the text and the exposition of conditions of validity and we consider now systems that are stationary in time, but not translation invariant in space. Only small adjustments of cosmetic nature have been made in the part on linear response. The conclusions have been adapted to account for the changes in the main part and some references have been added. Since the changes are big there is no point in listing them individually. Instead we provide a point-by-point reply to the comments of the referee.

1) Referee: "I would request the authors to give a broader introduction of their results. Namely they could specify how eq 1 was useful in Generalised Hydrodynamics and which physical implications it had for integrable systems."

Response: We would like to stress that this paper is not about integrable systems or GHD, but about a completely general property of stationary charge-current correlations. To avoid misunderstanding we have actually reduced the discussion of GHD and integrable systems which are just applications of recent interest. It is not within the scope of the present work to give an overview of these applications.

2) Referee: "It seems to me that it should be made more clear that in Generalised Hydrodynamics eq 1 is a consequence of local equilibrium and therefore not a priori valid at short times. The authors instead claim that it is valid at any time, especially after some quench from an inhomogenous initial state. Is that correct? If yes it should be made more clear in the introduction."

Response: We did not claim that Eq. (1) is valid immediately or any time soon after a quench. Instead, we pointed out that a GGE is reached ** asymptotically ** after a quench and ** then ** Eq. (1) can be applied. To avoid any misunderstanding along these lines, we have rewritten the discussion of the GGE.

3) Referee: "In the introduction they could mention what are the assumption for their proof. It seems to me that one request is that the state has sufficiently rapid decay of correlations, eq 28."

Response: We have completely restructured the derivation to make explicit in a systematic fashion which requirement leads to what. Translation invariance (in space) is shown to be not important and the role of decay of correlations is made explicit. (There are two distinct types of decay with different consequences.)

4) Referee: "below eq 34 they should define what is M and its physical meaning"

Response: Notation has been changed and the physical meaning of what formerly was denoted M has been made explicit (it is a density matrix that may not be of GGE form).

5) Referee: "slightly above eq 20: it seems that the authors want to point out that the symmetry eq 41 implies no hydrodynamics shocks. This point deserve to be expanded and made a bit more clear, also by referring to Generalised Hydrodynamics theory."

Response: We do not want to imply the absence of shocks. In fact, hyperbolic conservation laws can exhibit shocks. To judge whether shocks can appear one has to study the diffusive contribution to the hydrodynamic equation. This is beyond the scope of our work.

6) Referee: "in the conclusion they authors says that the correlation equality may be used as a probe of of the underlying asymptotic GGE, but it is not clear how."

Response: We have added an explanatory remark.

7) Referee: "in the conclusion the authors says that fluctuations are expected to be described by the KPZ class. It has been recently show [Phys. Rev. Lett. 121, 160603] that fluctuations are typically diffusive and in some cases super-diffusive, see Phys. Rev. Lett. 121, 230602 and arXiv:1812.02701. The authors may refer to these references maybe for their claim."

Response: We thank the referee for pointing out these two papers which are cited in the conclusions of the revised manuscript (plus two even more recent ones). We have also changed the wording to make clearer what we expect.

---

## Round 2 · Referee Report · Anonymous (Referee 2) · 2019-3-26

Strengths

  • short, simple discussion of certain interesting symmetry relations
  • generalised previous work to higher dimensions and find interesting consequences

Weaknesses

  • there is not a lot new that is actually done, relatively simple generalisations of known results
  • claims of novelty are too strong, it is the same symmetry relation as that found in the literature
  • citations to a fuller literature where such relations were found would be useful

Report

In this paper, the author derive and discuss a particular symmetry relation for conserved currents. The symmetry relation is a consequence of space-time stationarity and the conservation laws, as well as some conditions at infinity. The relation is derived in arbitrary dimension, and some interesting consequences on response functions are discussed.

This is a very short paper, concentrating on a specific relation. The discussion is interesting, putting together some bits and pieces found in the literature, with a slight generalisation to higher dimensions and some of their consequences. But adjustments are necessary before it can be published.

First, it is important to mention that the relation (1) (and (41)) was found before, in one dimension, with the same proof. I think the relation is not here established in the more general not-necessarily-commuting setting than in [1,2] or other papers before, see below. Maybe also already mention that the higher-dimensional version is obtained by projecting to one dimension.

Second, as emphasised in the derivations in literature, it should be made more clear what the assumptions are from the outset. On the first page the phrase "stationary invariant quantum system", and then later "translation invariance", is not too clear. What is required is that the state be space-time stationary, and the conservation laws hold, along with appropriate asymptotic conditions on correlation functions.

Third, I would also ask the authors to reduce the claims of surprise and novelty; for instance after equations 24,25, or the sentence "This result is remarkable as there is no a priori reason to expect any such general equality for correlations of physically unrelated conserved quantities and their currents." These are subjective, and surely for many people, these are not remarkable, consequences of known sum rules and mostly written in the literature already. I also have the feeling the way the paper is written sometimes "diminishes" the work done previously. For instance, the correlation equality in [4] is not a "specific correlation equality", it is the same equality as (1) (in one dimension); and I'm not sure why use the phrase "phenomenological hydrodynamics perspective" for the derivation in [4], as the same projection principles as those of [5] and of the Mazur bound are used (in all cases, I don't think "phenomenological" is the appropriate term).

Here are more explanations:

The relation (1), and its consequence (41), are relatively well known at least in some community. The authors cite [1] where a similar relation is derived in one dimension assuming commutativity of the stationary density matrix with the charges. They also cite [2] where it is derived in one dimension without this assumption (although [2] may be seen as almost simultaneous with the present paper). In both cases quantum systems were in mind. But the relation is also reviewed in [eq 2.30, H. Spohn, Fluctuating hydrodynamics approach to equilibrium time correlations for anharmonic chains], there in the case of three conserved currents but the statement makes it clear it is general. It is mentioned that the proof uses only space-time stationarity and the conservation laws, found in [H. Spohn, Nonlinear fluctuating hydrodynamics for anharmonic chains (2014)]. In these cases, classical systems were discussed, but the proof makes it clear that it does not depend on classicality. In fact, this type of relation, following from conservation laws and sum rules, have been studied for a long time, see for instance the book "Hydrodynamic fluctuations, broken symmetry, and correlation functions" (1975) by Dieter Forster.

The authors make the claim that the relation (1) they find, without using commutativity, is derived in [1] using commutativity, and it is different from that derived in [2] without commutativity because of the ordering of the operators on the right-hand side. They therefore claim that they have a relation derived in a more general setting in the quantum context. I believe this is incorrect. The relation is exactly the same as that from [2], no difference in operator order. See [eq B.5, 2], in both the left and right hand side, the indices keep the same order, exactly like in (1). It is in [1] that there is a different order of operators, which is reached by using commutativity: in [eq A1, 1], the first line of the derivation uses commutativity of the charge with the stationary density matrix. But skipping this first line, keeping the indices in the same order, one obtains (1). Hence, I would ask the authors to change this claim - in one dimension, it is the same relation as that found in the literature.

The derivation is also essentially the same as that in [1] or [2] or [H. Spohn, Nonlinear fluctuating hydrodynamics for anharmonic chains (2014)]: one uses space-time stationary and the conservation laws, in order to establish sum rules for quantities $D_i^{\alpha\beta}$. Hence also the derivation is not new, I think it would be useful for the readers if the authors could clarify this.

What is different from previous works is that here the relation is derived in higher dimensions. This is done by projecting to one dimension, integrating on the transverse components. Basically, once this projection is done, the derivation is the same as in one dimension.

Finally, small things:

  • After eq 35: it's not $M$ and $Q_\alpha$ not assumed to commute (as there is no $M$ in (35)), it is $Q_{\alpha}$ and $Q_{\alpha'}$

  • sum rules (24,25) are found in the literature, see [eq 2.21, 2] or [eq 2.29, H. Spohn, Fluctuating hydrodynamics approach to equilibrium time correlations for anharmonic chains].

Requested changes

See the report:

1- mention that the relation (1) (and (41)) was found before, in one dimension, with the same proof.

2- make more clear what the assumptions are from the outset.

3- reduce the claims of surprise and novelty; try to better represent work done previsouly; keep to the objective facts.

4- the small things mentioned in the report.

  • validity: good
  • significance: ok
  • originality: low
  • clarity: high
  • formatting: perfect
  • grammar: perfect

Author:  Gunter Schütz  on 2019-05-18  [id 522]

(in reply to Report 2 on 2019-03-26)

In response to the remarks by the referee, we have rewritten significantly the introduction and rewritten completely the main part on charge-current correlations. In particular, we consider now systems that are stationary in time, but not translation invariant in space. Only small adjustments of cosmetic nature have been made in the part on linear response. The conclusions have been adapted to account for the changes in the main part and some references have been added. Since the changes are big there is no point in listing them individually. Instead we provide a point-by-point reply to the comments of the referee.

1) Referee: "First, it is important to mention that the relation (1) (and (41)) was found before, in one dimension, with the same proof. I think the relation is not here established in the more general not-necessarily-commuting setting than in [1,2] or other papers before, see below. Maybe also already mention that the higher-dimensional version is obtained by projecting to one dimension."

Response: (a) Indeed, the relation (1) (and (41)) in the old version was found before, in one dimension, with the same proof as in the old version. The first mathematically rigorous proof of the relations (1) and (41) (old version) under the general assumptions of space-time invariance and decay of correlations is not due to Spohn, but was given, to the best of our knowledge, by Grisi and Sch\"utz in Ref. [22] (cited as [19] in the old version) which preceeds the work of Spohn by three years and which is cited in the papers by Spohn that the referee mentions. Under more specific conditions it was proved in 2003 by Toth and Valko which is also cited (Ref. [19] in the old version and [20] in the revised version) and which elaborates on the consequences of this relation for hydrodynamic limits. In order to clarify these issues we have pointed out in the revised version that [22] precedes [1] and [21].

(b) We thank the referee for pointing out a serious misreading on our part of the local relation of Ref. [6] (revised version). Indeed, we had confused the order of the operators and what we reported in our old version has the same ordering as Ref. [6]. To account for this in the revised version we have stated explicitly at the appropriate place (viz. where we introduce translation invariance) that we get the lattice analog of the local relation of Ref. [6]. In order to make the point that (1) and also its local analogue is nevertheless more generally valid than previously assumed ("assumption" in the sense of mathematical hypothesis used in the derivation, not in the sense of belief or expectation), we prove in the revised version analogous results, local and global, for systems that are stationary in time but not translation invariant in space. We also discuss in some more detail the finite-size corrections that appear in finite systems. Last but not least, unlike in Refs. [4,6] and in the old version of our manuscript we work now consistently on the lattice.

(c) As suggested by the referee, we have pointed out that the higher dimensional case is proved by projection (end of introduction) and we have explained how this is done by integrating out the perpendicular dimensions (end of Section II).

2) Referee: "Second, as emphasised in the derivations in literature, it should be made more clear what the assumptions are from the outset. On the first page the phrase "stationary invariant quantum system", and then later "translation invariance", is not too clear. What is required is that the state be space-time stationary, and the conservation laws hold, along with appropriate asymptotic conditions on correlation functions."

Response: We have completely restructured the derivation to make explicit in a systematic fashion which condition leads to what. Translation invariance (in space) is shown to be not important and the role of decay of correlations is made explicit. (There are two distinct types of decay with different consequences.)

3) Referee: "Third, I would also ask the authors to reduce the claims of surprise and novelty; for instance after equations 24,25, or the sentence "This result is remarkable as there is no a priori reason to expect any such general equality for correlations of physically unrelated conserved quantities and their currents." These are subjective, and surely for many people, these are not remarkable, consequences of known sum rules and mostly written in the literature already. I also have the feeling the way the paper is written sometimes "diminishes" the work done previously. For instance, the correlation equality in [4] is not a "specific correlation equality", it is the same equality as (1) (in one dimension); and I'm not sure why use the phrase "phenomenological hydrodynamics perspective" for the derivation in [4], as the same projection principles as those of [5] and of the Mazur bound are used (in all cases, I don't think "phenomenological" is the appropriate term)."

Response: The introduction has been rewritten significantly to make clear that this work is not about generalized hydrodynamics (which is just a field of application of strong recent interest), but about a completely general property of stationary charge-current correlations. In the new version the points that the referee makes (i.e. the question of "phenomenological", "specific" or "surprise") have become obsolete. Regarding novelty in comparison to the classical scenario: The generalization of (41) (which is (47) in the revised version) to the quantum case is in our opinion not immediate as one has to inspect the role of non-commutativity. For non-commutating charges, neither the present proof nor that of Refs. [2,4,6] works.

---

## Round 3 · Referee Report · Anonymous (Referee 3) · 2019-6-5

Strengths

quite exhaustive discussion
clear explanations, well organised
some results surprising

Weaknesses

none

Report

I think the authors have improved the manuscript greatly. I like how exhaustive it is, and how it makes clear the consequences of each assumptions. The results - in particular the relations obtained without space translation invariance - are quite interesting. This is a very nice work.

Requested changes

none

---

## Editorial Decision

published